Optic disc and cup segmentation for glaucoma detection using Attention U-Net incorporating residual mechanism

Chen Yuanyuan chenyuanyuan@lynu.edu.cn
Bai Yongpeng
Zhang Yifan
School of Information Technology, Luoyang Normal University , Luoyang , China
Pires Ivan Miguel
Electronic publication date: 2024 Mar 28
Publication date: 2024
Volume: 10
Electronic Location ID: e1941
Received 2023 Aug 20; Accepted 2024 Feb 26
Copyright: © 2024 Chen et al.
Copyright year: 2024
Copyright holder: Chen et al.
License: This is an open access article distributed under the terms of the Creative Commons Attribution License, which permits unrestricted use, distribution, reproduction and adaptation in any medium and for any purpose provided that it is properly attributed. For attribution, the original author(s), title, publication source (PeerJ Computer Science) and either DOI or URL of the article must be cited.
License URL: https://creativecommons.org/licenses/by/4.0/

Keywords: Glaucoma, U-Net, Residual module, Attention mechanism, Image segmentation

Funding: The authors received no funding for this work.

==============================
Glaucoma is a common eye disease that can cause blindness. Accurate detection of the optic disc and cup disc is crucial for glaucoma diagnosis. Algorithm models based on artificial intelligence can assist doctors in improving detection performance. In this article, U-Net is used as the backbone network, and the attention and residual modules are integrated to construct an end-to-end convolutional neural network model for optic disc and cup disc segmentation. The U-Net backbone is used to infer the basic position information of optic disc and cup disc, the attention module enhances the model’s ability to represent and extract features of optic disc and cup disc, and the residual module alleviates gradient disappearance or explosion that may occur during feature representation of the neural network. The proposed model is trained and tested on the DRISHTI-GS1 dataset. Results show that compared with the original U-Net method, our model can more effectively separate optic disc and cup disc in terms of overlap error, sensitivity, and specificity.

Introduction

Glaucoma is one of the main causes of blindness in China, and its most serious consequence is visual function damage that is irreversible. In general, glaucoma cannot be effectively prevented, but if it is detected early and treated reasonably in time, the visual function of most patients can be maintained effectively for life. Therefore, the prevention and treatment of glaucoma must place emphasis on early detection, diagnosis, and treatment (Ahmed et al., 2015; Quigley & Broman, 2006). Glaucoma is usually diagnosed by professional physicians through observation, while this approach is inefficient and inaccurate (Lim et al., 2015). In clinical diagnosis, glaucoma is usually diagnosed from the following aspects: cup disc vertical diameter ratio (CDR) and cup disc area ratio (RDAR). Between them, CDR (Akram et al., 2015) is accepted by most doctors, the higher the CDR value is, the greater the risk of glaucoma. 3D optical coherence tomography (OCT) is currently available for CDR measurements, but reaching most areas by using this method is too costly, so fundus imaging remains the diagnostic method for most doctors (Agrawal et al., 2018). With the deepened integration of deep learning and the medical field, people began using deep learning methods for glaucoma diagnosis, and these methods are of great value to glaucoma patients, especially in areas lacking ophthalmologists (Bastawrous et al., 2015; Lodhia et al., 2016). The premise of diagnosis is to divide the area of the optic disc (optic disc) and cup disc for the calculation of CDR indicators, so how to accurately segment optic disc and cup disc areas from the fundus image has become the key to glaucoma diagnosis.

With the development of computer vision, image processing has been applied in the medical image field. In medical image analysis, the quality of images directly affects the design and accuracy of recognition algorithms. Therefore, image preprocessing is necessary before conducting image analysis such as feature extraction, segmentation, matching, and recognition. The main purpose of image preprocessing is to eliminate irrelevant information in the image, restore useful real information, enhance the detectability of target feature information, and simplify the data to the greatest extent, thereby Proper image processing can enhance the feature information required in the image, reduce the influencing factors, and enable the computer to further understand the image to achieve a similar understanding of the image as the human visual system.

Image segmentation can locates the boundaries and contours of the target in the form of regions, and a machine must understand the image. Medical image segmentation mainly deals with various image segmentation problems involved in the medical field, such as common magnetic resonance imaging (MRI) scanning images. Its main task is to segment regions of interest from these medical images, such as specific organ parts and interest targets (such as glaucoma). Unlike common segmentation tasks in daily life, medical images (such as MRI images) may encounter problems such as low contrast, low signal-to-noise ratio, and low light intensity due to the influence of image acquisition equipment; the organs themselves undergo movement and deformation (such as the heart), and there are also differences between individuals. These factors have led to the difficulty of medical image segmentation and its own algorithmic design characteristics.

With the rise of artificial intelligence, image segmentation based on deep learning has made great progress, and relevant researchers have proposed a series of neural network models such as AlexNet (Krizhevsky, Sutskever & Hinton, 2012), VGG (Simonyan & Zisserman, 2014), and ResNet (He et al., 2016). Traditional machine learning is sensitive to the changing noise and complexity of images. Deep learning, especially U-Net-based neural network models, has achieved enhanced performance in the medical images with small data volume, low image quality, diversified modality, and complex structure. Furthermore, U-Net-based neural network models are widely used in medical image segmentation.

Fundus image segmentation is a multiclassification image segmentation problem. All pixels of an image must be divided into three categories: background, optic disc, and cup disc. The cup disc area is on the side of the disk area, as shown in Fig. 1. However, in an entire fundus image, optic disc and cup disc trays mainly occupy a small area, so the image usually needs to be pre-processed before segmentation. In the usual preprocessing method, the approximate area of the disc is predicted to reduce the training range and improve the accuracy. At present, the two main strategies of splitting optic disc and cup disc are to divide them separately and jointly. Image segmentation based on deep learning can better complete feature extraction in accordance with the provided label than segmentation methods that are based on traditional machine learning, thus greatly improving efficiency. Recently, these models have received much attention and been well developed in medical image segmentation. Relevant researchers have proposed many neural network models such as the M-Net (Fu et al., 2018) based on U-Net (Ronneberger, Fischer & Brox, 2015) proposed by Fu et al. (2018), the neural network based on modified U-Net proposed by Sevastopolsky (2017), and the neural network based on U-Net++ (Tulsani, Kumar & Pathan, 2021).

Figure 1 Optic disc and cup disc (the one circled in red is the optic disc, while the one in yellow is the cup disc).

Accurate detection of optic disc and cups disc plays a key role in glaucoma diagnosis. A new end-to-end deep learning model was constructed in this article to alleviate the difficulty of the previous algorithm model in effectively segmenting optic disc and cup disc joints. The network was built on U-Net, which does not require complex preprocessing of images to achieve good prediction effect. In this article, the segmentation of optic disc and cup disc was described as a multilabel classification problem, which reduces the complexity of the manual task. The results have high application value, and they could greatly reduce the cost of glaucoma detection in remote areas with limited medical resources.

Related work

Optic disc segmentation

The retina from the macula, which is about 3 mm to the nasal side, has a reddish disc-like structure with a diameter of about 1.5 mm and a clear boundary, called the optic nerve disc or the optic nerve papilla and referred to as optic disc in this article. Before deep learning, handmade feature models or templates were used for optic disc detection. For example, the method proposed by Zhu & Rangayyan (2008) was applied to automatically position the optic disc in retinal fundus images by using the Canny or Sobel method for edge detection and the hough transform for circle detection. The template-based method presented by Aquino, Gegundez-Arias & Marin (2010) uses morphological and edge detection techniques to obtain circular optic disc boundary approximation through the circular Hough transform. Numerous individuals have begun employing deep learning-driven segmentation using deep learning techniques due to the vulnerability of manually crafted images to low quality. For instance, the unified retinal image analysis framework developed by Maninis et al. (2016) and team, known as deep retinal image understanding, utilizes VGGnet (Simonyan & Zisserman, 2014) and transfer learning techniques of deep convolutional neural networks to provide retinal vascular and optic disc segmentation. This approach has yielded impressive results, particularly in cup disc segmentation.

Cup disc segmentation

The cup disc is a normal physiological depression inside the optic disc. The contrast between the cup disc boundaries in the 2D fundus image is very low, which gives a challenging for dividing the cup disc. Some cup disc segmentation directly follows the method of optic disc segmentation. Only are changes made to the feature model, and certain effects are obtained. However, the accuracy and performance did not greatly be improved. When entering the cup disc from optic disc is the curvature of the blood vein, the cup disc has an obvious feature, therefore, some traditional methods use this feature to divide the cup disc. With the advent of deep learning and the problem that handmade features may not be applicable in low-quality images or alternative datasets, Zilly, Buhmann & Mahapatra (2015) proposed a new optic nerve cup and optic disc segmentation method that is based on convolutional neural networks. This method uses a two-layer multiscale convolutional neural network trained by boosting. Entropy-based sampling was introduced to reduce the computational complexity, and its results are superior to those of uniform sampling. The evaluation results on the DRISHTI-GS database showed that the IoU and Dice scores were superior to those of the original methods. However, this method needs to crop the image in accordance with the optic disc area before segmenting the optic disc, so the bounding box of optic disc and cup disc must be prepared in advance. Therefore, such method is not suitable for full-fundus images that have not been seen before. Later, an improvement of the above method in the training process of convolutional filter was proposed (Zilly, Buhmann & Mahapatra, 2017). It uses entropy sampling to select information points. Entropy sampling reduces the computational complexity, and it is better than uniform sampling. In this method, a new boost-based convolutional filter learning framework was designed using sampling points. The filter learns through multiple layers, with the output of the previous layer serving as the input to the next layer. A softmax classifier is then trained on the output of all learning filters and applied to the test image. The output of the classifier is obtained by the unsupervised graph cutting algorithm and the convex hull transformation to obtain the final segmentation result (Guo et al., 2019). The experimental results on DRISHTI-GS and RIM-ONE databases (version 3) show that compared with the previous method, this method does not need to be cropped through the light cup region when segmenting images, so it has strong applicability.

Joint segmentation of optic disc and cup disc

Joint segmentation of optic disc and optic cup (OC) has also received attention. Sevastopolsky (2017) proposed a deep learning-based universal method for automatic segmentation of optic disc and OC in 2017. The modified U-Net neural network consists of an encoding path (left) and a decoding path (right). The proposed modification has fewer filters in all convolutional layers than the original U-Net, and it does not increase the number of filters used to increase samples. These changes do not reduce the recognition quality of the task and make the architecture more lightweight in terms of parameter number and training time decrease. Meng et al. (2022) proposed a semi-supervised framework combining PM, MSDF, and B-ROI to solve the problem of training time and cost, and embedded vCDR computation. This framework has higher vCDR computational efficiency and better segmentation performance than fully supervised segmentation. M-Net is another neural network model that is based on U-Net proposed by Fu et al. (2018) and his team in 2018. It simultaneously solves the segmentation problem of optic disc and OC in a single-stage multilabel system. M-Net mainly consists of a multiscale input layer, a U-shaped convolutional network, side output layers, and a multilabel loss function. The multiscale input layer constructs an image pyramid to receive data of different sizes at multiple scales. M-Net also uses U-Net as the main network structure to learn rich hierarchical representations, and the side output layers serve as early classifiers to generate corresponding local prediction maps for different scales. Then, a multilabel loss function is proposed to generate the final segmentation map. Polar coordinate transformation is introduced to further improve the segmentation performance of M-Net. After polar coordinate transformation, the relationship between optic disc and the cup disc, which is contained in the original image, becomes the relationship between layers in polar coordinates. This phenomenon not only allows segmentation by layer separation but also makes the ratio of the cup disc to the optic disc more balanced, which can effectively prevent overfitting and further improve the segmentation performance. Haider et al. (2023) proposed FBSS-Net on the basis of efficient shallow segmentation network (ESS-Net). FBSS-Net used MDSP in ESS-Net to preserve deep semantic information. The authors also proposed a mixed structure of internal and external features to further improve the overall segmentation performance. Yi et al. (2023) designed a primitive Coarse-to-Fine Transformer Network (C2FTFNet) to jointly segment optic disc and OC to address the complexity of clinical data. C2FTFNet not only alleviates the limitations of traditional convolution methods on receptive fields but also reduces the semantic gap between different hierarchical features. Zhang et al. (2023) proposed a novel and general distance-guided deep learning strategy (DGLS) to simultaneously segment optic disc and OC from color fundus images on the basis of available networks (i.e., U-Net). The algorithm exhibits superior performance compared with U-Net and several variants trained on a single annotation and traditional training strategies. Wang, Li & Cheng (2023) proposed an extended EfficientNet-based U-Net called EE-U-Net to improve the segmentation accuracy. Quantitative and qualitative results validated the superior performance of EE-U-Net in OC and optic disc segmentation, demonstrating that it was a promising tool for early screening of large-scale glaucoma. Bhattacharya et al. (2023) proposed a robust segmentation pipeline for optic disc and OC segmentation utilizing the U-Net architecture. This model outperforms most advanced algorithms available for disc and cup segmentation tasks. Moris et al. (2023), Guo et al. (2019) conducted a comprehensive analysis of different coarse-to-fine designs of optic disc/OC segmentation from the perspective of standard segmentation and vCDR for evaluating glaucoma. The analysis showed that when these methods learn from particularly large and diverse training sets, they do not outperform the standard multiclass single-level models (Guo et al., 2019).

Model

Accurate detection of the optic disc and cup disc plays a crucial role in the diagnosis of glaucoma. In this article, an end-to-end convolutional neural network model for glaucoma detection was constructed using U-Net as the backbone network and by integrating the attention module (convolutional block attention module, abbreviated as CBAM) (Woo et al., 2018) and residual module. U-Net was used as a backbone to infer the basic position information of optic disc and the cup disc. CBAM help the model locate circular or boundary optic disc features or blood vessel bending features. It not only effectively enhanced the model’s representation and extraction ability for optic disc or cup disc features but also greatly help improve the performance via the accurate detection of optic disc and cup disc. The residual module effectively alleviates the problem of gradient disappearance or gradient explosion that may occur in the process of learning feature representation. So, the network model can more accurately detect and classify patients with and without glaucoma.

Overall framework

In this article, a glaucoma detection segmentation network model for optic disc and cup disc based on U-Net was proposed. As shown in Fig. 2, the overall network structure consists of two parts: encoding network and decoding network (the specific network parameters of each layer are shown in Table 1). The entire network structure resembles the letter “U,” similar to the original U-Net network structure, which consists of a shrink path (left) and an expansion path (right). The persistent connection in the U-shaped network model helps restore the information loss caused by continuous down sampling. The left part of the network model is the encoding process, whose purpose is to perform feature extraction to enhance the ability to extract the features of the cup disc and optic disc. CBAM was added to the encoding process. A residual module was also added to the encoding process to prevent vanishing gradients and bursting gradients. The right part of the network is the decoding process, which aims to restore image information and feature fusion. The transformed network is still an end-to-end network, which only uses one model and one objective function. It has the advantages of reducing the complexity of the project and avoiding the defects of error accumulation caused by the use of different modules and objective functions.

Figure 2 U-Net+Res+CBAM network structure diagram.

Table 1 The parameters of proposed model.

Layer	Input	Output	Kerenl	Stride	Padding	Activation	
Conv1_1	3 × 640 × 640	64 × 640 × 640	3 × 3	1	1	LeakRelU	
Maxpool2d	64 × 640 × 640	64 × 320 × 320	3 × 3	2	1	LeakRelU	
Conv2_1	64 × 320 × 320	128 × 320 × 320	3 × 3	1	1	LeakRelU	
Maxpool2d	128 × 320 × 320	128 × 160 × 160	3 × 3	2	1	LeakRelU	
Conv3_1	128 × 160 × 160	256 × 160 × 160	3 × 3	1	1	LeakRelU	
Maxpool2d	256 × 160 × 160	256 × 80 × 80	3 × 3	2	1	LeakRelU	
Conv4_1	256 × 80 × 80	512 × 80 × 80	3 × 3	1	1	LeakRelU	
Maxpool2d	512 × 80 × 80	512 × 40 × 40	3 × 3	2	1	LeakRelU	
Conv5_1	512 × 40 × 40	1,024 × 40 × 40	3 × 3	1	1	LeakRelU	
Upsampling	1,024 × 40 × 40	1,024 × 80 × 80	1 × 1	1			
Conv6_1	1,024 × 80 × 80	512 × 80 × 80	3 × 3	1	1	LeakRelU	
Conv6_2	512 × 80 × 80	512 × 80 × 80	3 × 3	1	1	LeakRelU	
Upsampling	512 × 80 × 80	512 × 160 × 160	1 × 1	1			
Conv7_1	512 × 160 × 160	256 × 160 × 160	3 × 3	1	1	LeakRelU	
Conv7_2	256 × 160 × 160	256 × 160 × 160	3 × 3	1	1	LeakRelU	
Upsampling	256 × 160 × 160	256 × 320 × 320	1 × 1	1			
Conv8_1	256 × 320 × 320	128 × 320 × 320	3 × 3	1	1	LeakRelU	
Conv8_2	128 × 320 × 320	128 × 320 × 320	3 × 3	1	1	LeakRelU	
Upsampling	128 × 320 × 320	128 × 640 × 640	1 × 1	1			
Conv9_1	128 × 640 × 640	64 × 640 × 640	3 × 3	1	1	LeakRelU	
Conv9_2	64 × 640 × 640	64 × 640 × 640	3 × 3	1	1	LeakRelU	
Conv10_1	64 × 640 × 640	3 × 640 × 640	1 × 1	1	1	Sigmoid	

Feature extraction

The left part of the modified U-Net network is the feature extraction part. After the original image is supplemented into a square according to the longer side and then reduced in equal scale (such a picture preprocessing workload is significantly reduced compared to most other models), the image resolution is 640 × 640, and the number of channels is 3. After a 3 × 3 convolution operation, the resolution remains unchanged, and then, the number of channels increases from 3 to 64, which aim is mainly to increase sampling and achieve the effect of increasing feature information. In the process of convolution, the number of channels is not directly changed by the convolution function, but the residual network is directly introduced, and the fused operation of convolution function and the residual network is used to change the number of channels. The final size is 40 × 40 × 1,024 feature blocks (the specific process is shown in Fig. 3).

Figure 3 The process of extracting features.

Residual network

On the one hand, using a residual network can better simulate the classification function to obtain higher classification accuracy. On the other hand, it can alleviate gradient vanishing and gradient explosion problems. ResNet (He et al., 2016) demonstrates a better model structure by changing the model structure, that is, changing the shape of the error surface.

ResNet calls the stacked layers a block, and the function that each block can fit is F(x), which is assumped the desired latent map is F(x), instead of letting F(x) learn the latent map directly, the residual H(x)−x is learned, In detail, F(x):=H(x)−x. The forward path becomes F(x)+x, and F(x)+x is used to fit H(x). F(x) can more easily learn to be 0 than for F(x) to learn identity maps, which can be easily achieved by L2 regularity. In this manner, for redundant blocks, only F(x)→0 can be used to obtain an identity map without loss of performance. The block formed by F(x)+x is called a residual block (Fig. 4).

Figure 4 Residual block module.

In Fig. 4, the first weight layer performs dimensionality reduction by convolving with the input image to reduce computational complexity. After passing through the activation function, the second weight layer performs another convolution to increase dimensionality and ensure that the output image has the same dimensions as the input image to facilitate subsequent addition operations, where corresponding elements in the feature map are added together and passed through an activation function to produce the final output. When the identity mapping is optimal, the extracted features of the previous layer is optimal. The identity map at this time is the underlying map, i.e., H(X)=X. However, in most cases, the identity map cannot be optimal, so the residual module needs to be corrected.

Attention mechanism

The feature map obtained after the residual operation is then fed into CBAM, which is a simple and effective attention mechanism module proposed by Yi et al. (2023). Integrating this module into the encoding process of the U-Net network not only enhances the network model’s feature extraction ability but also significantly improves the network model’s performance by learning feature representation of spatial and channel information in an image.

CBAM is a mixed-domain attention mechanism that includes channel and spatial domains. Different dimensions represent different meanings and carry different information. In this article, the channel is the overall semantic expression of the optic disc and cup disc feature abstraction. Meanwhile, the space has richer position feature information of the cup disc and optic disc. CBAM integrates two different attention modules in a serial fashion: the channel attention module and the spatial attention module. Given the feature map F containing the middle features of optic disc and the cup disc as input, CBAM infers the one-dimensional channel attention image (overall prediction) and the two-dimensional spatial attention image (that is, the specific position prediction of the feature information of optic disc and the cup disc) in turn. The channel attention feature map corrects the original feature map F to obtain F′, and the spatial attention feature map corrects the feature map F′ again to obtain F′′. After one CBAM, the final output of the feature map F′′ has the same resolution as the input feature map F image resolution and the same number of channels. The entire process is summarized in Fig. 5.

Figure 5 Convolutional block attention module.

(1) F′=MC(F)⊗F,F′′=MS(F′)⊗F′, ⊗ representing the multiplication of elements 

Details of each attention module are described below. For channel attention module, given an input F∈RH∗W∗C, the spatial information of the feature map is aggregated through global average pooling (GAP) and global maximum pooling (GMP) operations, and different spatial semantic description operators FCavg and FCmax, respectively, are obtained. The two are fused by a shared perceptron, and then the two channel attention feature vectors are fused using an additive method. Finally, after the sigmoid activation function is used, the channel domain attention vector MC, MC∈RC∗1∗1 is obtained. The detailed description is shown in Fig. 6.

Figure 6 Channel attention module.

(2) MC(F)=σ(MLP(AvgPool(F))+MLP(MaxPool(F)))=σ(W1(W0(Favgc))+W1(W0(Fmaxc))),

where σ represents the sigmoid activation function, W1∈RC×C/r and W0∈RC/r×C. The MLP weights W0 and W1 are shared for both inputs, and the ReLU activation function is followed by W0.

For spatial attention module, given an input F∈RH∗W∗C, dimensionality reduction is applied along the channel dimension through GPA and GMP operations, which not only reduce a large number of operations but also effectively highlight the information region. Afterwards, two different channel characterization operators FSavg and FSmax are obtained, and the two are spliced to change the channel to 2. Then, a convolution operation with a convolution kernel of 7 × 7 is used to achieve dimensionality reduction and expand the receptive field. The expansion of the receptive field is important for processing the spatial information of cups and discs. It also improves the classification accuracy. After the sigmoid activation function, the spatial domain attention vector MS, MS∈R1∗H∗W is obtained, and the detailed description is shown in Fig. 7.

Figure 7 Spatial attention module.

(3) MS(F)=σ(f7×7([AvgPool(F);MaxPool(F)]))=σ(f7×7([FavgS;FmaxS])),

where σ represents the sigmoid activation function, f7×7 represents a convolution operation with a convolution kernel of 7 × 7, AvgPool represents GAP operation, and MaxPool represents a GMP operation.

After the above CBAM processing is used, the number of image channels and resolution remain unchanged, and a 640 × 640 × 64 image is obtained, which is recorded as step 1. Next, step 1 undergoes a down-sampling operation (achieved by 2 × 2 maximum pooling), and the number of channels remains unchanged in the operation step. The resolution size changes from 640 × 640 to 320 × 320. After down-sampling, the residual module is entered, the resolution size of the image remains unchanged, and the number of channels is doubled from 64 to 128. Then, through CBAM, the number of channels and resolution are unchanged, and a 320 × 320 × 128 image is obtained, which is recorded as step 2. The down-sampling operation is repeated, where the number of channels remains unchanged at 128, whereas the resolution size changes from 320 × 320 to 160 × 160. After passing through the residual module, the number of channels is doubled from 128 to 256. Then, through CBAM, the number of channels and resolution are unchanged, and a 160 × 160 × 256 image is obtained, which is denoted as step 3. The down-sampling, residual module, and CBAM operations are repeated to obtain an image of 80 × 80 × 512, which is recorded as step 4. Then, they are repeated to finally obtain a 40 × 40 × 1,024 image, which is denoted as step 5, which is the end of the encoding process.

Decoding process

The right part of the transformed U-Net network is the feature fusion and restoration part, which is also composed of four layers (Fig. 8). Starting from the feature map step 5, it enters the expansion path and is up-sampled (implemented through deconvolution) before decoding, with the channel number halving from 1,024 to 512, and the image size doubling from 40 × 40 to 80 × 80. The obtained image is then concatenated with the corresponding feature map step 4 of the left shrinking path of the same size, and the channel number is doubled from 512 to 1,024 after concatenation. After two 3 × 3 convolutions, the first convolution operation halves the number of channels of the feature map from 1,024 to 512, and the image size remains unchanged. After the second convolution, the number of channels and size of the feature map remain unchanged at 80 × 80 × 512. Each convolution is followed by BN processing to reposition the convoluted result in a certain distribution, followed by regularization to mitigate overfitting. Finally, the convolutions are processed by activation function. The BN layer can speed up training and improve the generalization ability of the network. Regularization can reduce the model’s reliance on local features. Each layer in the extension path is the same network described above. After this layer, the feature map is up-sampled and then concatenated with the corresponding feature map step 3 on the left side. After two convolutions, the size of the feature map becomes 160 × 160 × 256. Then, the two layers of such a network are repeated to obtain a feature map with a size of 320 × 320 × 64, and convolution and Sigmoid operation are conducted on the graph. The number of channels is set in accordance with the number of classifications. Finally, a prediction map of 640 × 640 × 3 is outputted, at which point the decoding process ends.

Figure 8 The process of decoding features.

Experiment and result analysis

Datasets

This experiment used the DRISHIT-GS (Sivaswamy et al., 2014, 2015) dataset provided by the Medical Image Processing (MIP) group, IIIT Hyderabad. This dataset is a standard dataset for glaucoma optic disc and OC segmentation. Compared with other datasets that only provide the original image, the DRISHTI-GS1 dataset provides multiple experts for the diagnosis of glaucoma in the original image, manual segmentation of optic disc and the cup disc, and the corresponding evaluation indicators. It is more standardized than other datasets, and it reduces the doorsill for splitting optic disc and OC. The DRISHTI-GS1 dataset contains 101 images in PNG format, and the resolution is not uniform. The transformed network requires the input resolution to be 640 × 640, so the image needs to be processed simply before entering it. First, the picture is supplemented by the longer side to a square size and then scaled to 640 × 640 in equal proportions (Fig. 9). This experiment used the average boundary provided by the experts in the dataset as the label, which needs to be converted to an image because the boundary provided is in text form (Fig. 10).

Figure 9 Image preprocessing.

Figure 10 Label preprocessing.

Evaluation indicators

In this experiment, sensitivity, specificity, and overlapping errors were used as evaluation indices, and their calculation formulas are as follows:

(4) OverlappingError:E=1−Aera(S∩G)Aera(S∪G),

(5) Sensitivity:SEN=TPTP+FN,

(6) Specificity:SPE=TNTN+FP,

where S is the predicted result, and G is the corresponding ground truth. The TP, TN, FP, and FN indicators in the above formulas serve as follows:

TP indicates that the pixel is an optic disc (cup disc) element in the label, and if it is successfully predicted as an optic disc (cup disc) element in the prediction graph, the pixel is marked as TP point.

TN indicates that the pixel is a non-optic disc (non-cup disc) element in the label, and if it is successfully marked as a non-optic disc (non-cup disc) element in the prediction graph, the pixel is marked as TN point.

FP indicates that the pixel is a non-optic disc (non-cup disc) element in the label, and if it is predicted as an optic disc (cup disc) element in the prediction graph, the pixel is marked as an FP point.

FN indicates that the pixel is an optic disc (cup disc) element in the Label, and if it is predicted as a non-optic disc (non-cup disc) element in the prediction graph, the pixel is marked as FN point.

In this experiment, the overlapping error rate (optic disc_E), sensitivity (optic disc_SEN), and specificity (optic disc_SPE) of optic disc and the overlapping error rate (Cup_E), sensitivity (Cup_SEN), and specificity of the cup disc (Cup_SPE) were calculated.

Experimental platform

This experiment ran on the Ubuntu operating system and used python 3.6 and Pytorch for code writing and building neural networks. The running environment uses GPU to speed up image processing, the graphics card is NVIDIA 3090, and the CUDA version is 11.3.

Experiment setting

In this experiment, Adam optimizer was adopted, which can not only replace the traditional stochastic gradient descent algorithm but also automatically adjust the learning rate to optimize the model parameters, with a faster convergence speed. The batch size set in the training phase is 3, and the training epoch is 900 epochs.

Loss function

After the loss function BCE is calculated, it is divided by the number of points and finally returns to the average loss. So, this loss function can handle the multiclassification problem. The aimed at the joint segmentation of fundus images, which belongs to the multilabel multiclassification problem. Therefore, the loss used in this article is BCE, and the loss function is defined as follows:

(7) L(p,t)=−(∑in⁡(t×log(p)+(1−t)×log(1−p)))/n,

where t represents the ground true value, and p is the corresponding predicted value.

Effectiveness analysis of fusion attention mechanisms and residual modules

Two comparative experiments were designed, and the effectiveness of U-Net improvement was demonstrated through pictures and data. The first comparative experiment was conducted between the original U-Net and the improved U-Net network, demonstrating the effectiveness of CBAM and the residual module. In (1) and (2), visual and data analyses were performed separately for the first comparative experiment. On the second experiment, U-Net networks that underwent different degrees of modification were compared, with one model only modified in the encoding phase and the other model modifying the encoding and decoding stages. This experiment aimed to demonstrate that the CBAM and residual modules not only effectively enhance the feature extraction capability in the encoding phase but also improve the feature fusion capability in the decoding phase. The results of the second comparative experiment are presented in the “Effectiveness analysis of fusion attention mechanisms and residual modules” section by comparing pictures and data.

(1) Visual analysis

Figure 11 compares the original U-Net with the improved U-Net test results to show the effectiveness of the CBAM and residual modules more intuitively. The feature extraction ability of the network improved by replacing the convolutional layer of U-Net with the CBAM and residual modules, and the test results of the improved model are closer to label than the original U-Net, proving that both modules are effective. These results reflect that integrating CBAM with residual modules allows the network to focus on important features by adaptively recalibrating the feature maps in different spatial locations and channels. This attention mechanism helps the proposed network to learn more discriminative features, leading to improved performance in our segmentation tasks.

Figure 11 Partial visualization on the DRISHTI-GS1 dataset: (A) original image, (B) label, (C) U-Net, (D) U-Net+CBAM+residual.

(2) Analysis under different evaluation indicators

In addition to visualizing and comparing the test results, the effectiveness of the CBAM and residual modules was proven by calculating the three evaluation indicators of overlapping error, sensitivity, and specificity. Overlapping error represents the similarity of test results and label, sensitivity represents the degree of prediction of optic disc or cup and disc area, and specificity represents the degree of prediction of non-optic disc or the non-cup and disc area. The lower the overlapping error is, the higher the sensitivity and specificity represent, and the better the performance of the model. As shown in Tables 2 and 3, the segmentation capabilities of the two U-Net models and three models that are not based on U-Net (Gao et al., 2020; Son, Park & Jung, 2019; Gu et al., 2019) against optic disc and the cup disc were compared, and the results further proved that the CBAM and residual modules can improve the feature extraction ability of the model and its segmentation performance. The above content may illustrate that CBAM’s attention modules allow for adaptive feature refinement, enabling the network to dynamically adjust the importance of different features. When integrated with residual modules, this adaptive feature refinement can lead to more effective utilization of the network’s capacity, resulting in improved representation learning and model performance.

Table 2 Comparison results of U-Net and the proposed model on optic disk data.

Model	OD_SEN	OD_SPE	OD_E	
U-Net	0.9887	0.9943	0.1179	
U-Net+CBAM+residual	0.9974	0.9967	0.1075	
RecurrentUnits (Gao et al., 2020)	0.9374	---	---	
GAN (Son, Park & Jung, 2019)	0.9747	0.9977	---	
CE-Net (Gu et al., 2019)	0.9759	0.9990	---	

Table 3 Comparison results of U-Net and the proposed model on cup and plate data.

Model	Cup_SEN	Cup_SPE	Cup_E	
U-Net	0.9084	0.9963	0.3387	
U-Net+CBAM+residual	0.9559	0.9971	0.3114	
RecurrentUnits (Gao et al., 2020)	0.9533	---	---	
GAN (Son, Park & Jung, 2019)	0.8539	0.9907	---	
CE-Net (Gu et al., 2019)	0.8819	0.9909	---	

(3) Effectiveness analysis of adding residual modules in the encoding stage and adding residual modules in the whole process

Experiments were conducted by adding residual modules separately in the encoding and the whole process (encoding and decoding stages) to verify the importance of adding CBAM and the residual module at different positions. The addition of CBAM and residual modules at different locations is shown in Fig. 12. The prediction results of adding residual networks in the whole process are smoother at the boundary than adding residual networks only in the encoding stage. This finding verified the promoting effect of adding residual networks in the whole process on segmentation performance. Tables 4 and 5 show the experimental results of overlapping error, sensitivity, and specificity on optic disc and cup disc data after adding the residual network at different locations. The results of the two U-Net models are close, but the addition of the residual network in the whole process is still slightly better than the model structure of the residual network only added in the encoding stage. Compared with other three models that are not based on U-Net, the results of sensitivity and specificity in optic disc segmentation were closely matched. Additionally, in the sensitivity of cup segmentation, the proposed model outperforms other methods. Moreover, these results also illustrate to us a fact of application: residual modules facilitate the training of very deep neural networks by mitigating the vanishing gradient problem. This enables the model to learn more effectively from the training data, allowing for the successful training of deeper architectures without suffering from degradation in performance.

Figure 12 Visualization comparison of the addition of residual modules at different stages: (A) original image, (B) label, (C) encoder, (D) encoder & decoder.

Table 4 Comparison results of adding residual modules at different stages on the optic disc data.

Stage	OD_SEN	OD_SPE	OD_E	
Encoder	0.9912	0.9951	0.1125	
Encoder & decoder	0.9974	0.9967	0.1075	
RecurrentUnits (Gao et al., 2020)	0.9374	---	---	
GAN (Son, Park & Jung, 2019)	0.9747	0.9977	---	
CE-Net (Gu et al., 2019)	0.9759	0.9990	---	

Table 5 Comparison results of adding residual modules at different stages on the cup disc data.

Stage	Cup_SEN	Cup_SPE	Cup_E	
Encoder	0.9296	0.9965	0.3276	
Encoder & decoder	0.9559	0.9971	0.3114	
RecurrentUnits (Gao et al., 2020)	0.9533	---	---	
GAN (Son, Park & Jung, 2019)	0.8539	0.9907	---	
CE-Net (Gu et al., 2019)	0.8819	0.9909	---	

Conclusion

Glaucoma is currently the second leading cause of blindness among eye diseases, so its prevention is very important for eye health. However, CDR values are an important indicator for glaucoma diagnosis, and the prerequisite for calculating this indicator is identifying the area of optic disc and the cup disc. However, the costs of conventional manual diagnosis methods and automatic diagnosis are considerably high, so automatically segmenting optic disc and the cup disc from the lower-cost fundus image to calculate the CDR value has become a popular topic in deep learning. Based on U-Net deep learning neural network, a deep learning neural network for fundus image optic disc and OC segmentation was established by fusing the residual module and attention mechanism. The main contributions are as follows: reducing the workload of picture and processing, and realizing the joint segmentation of optic disc and the cup disc. After the DRISHTI-GS1 dataset was tested, the accuracy of the modified neural network was found to be better than that of the original U-Net structure. The results play a very important role in reducing the cost of glaucoma diagnosis. Such a reduction means that the diagnosis of glaucoma can not only be popularized in more economically underdeveloped areas but also help solve the problem of uneven medical resources.

In the future, how to effectively solve the problems of cup disc segmentation and fuzzy image segmentation will be the research direction.

Supplemental Information

Supplemental Information 1 Dataset.

Supplemental Information 2 code.

Additional Information and Declarations

Competing Interests

Author Contributions

Data Availability

The authors declare that they have no competing interests.

Yuanyuan Chen conceived and designed the experiments, prepared figures and/or tables, authored or reviewed drafts of the article, original write, and approved the final draft.

Yongpeng Bai conceived and designed the experiments, performed the experiments, performed the computation work, prepared figures and/or tables, authored or reviewed drafts of the article, and approved the final draft.

Yifan Zhang conceived and designed the experiments, analyzed the data, performed the computation work, prepared figures and/or tables, authored or reviewed drafts of the article, and approved the final draft.

The following information was supplied regarding data availability:

The DRISHTI-GS1 dataset is available at http://cvit.iiit.ac.in/projects/mip/drishti-gs/mip-dataset2/Home.php.

The code is available at GitHub and Zenodo:

- https://github.com/Kid-FanFan/CupAndDisc_Segmentation.

- Yuanyuan, C., Yongpeng, B., & Yifan, Z. (2023). CupAndDisc_Segmentation. Zenodo. https://doi.org/10.5281/zenodo.10012225.

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
