# Peer review of "Optic disc and cup segmentation for glaucoma detection using Attention U-Net incorporating residual mechanism"

_PeerJ Computer Science, doi:10.7717/peerj-cs.1941_

## Round 0.1 · original submission · Major Revisions

Please consider the reviewers' comments.

**Language Note:** The review process has identified that the English language must be improved. PeerJ can provide language editing services - please contact us at [email protected] for pricing (be sure to provide your manuscript number and title). Alternatively, you should make your own arrangements to improve the language quality and provide details in your response letter. – PeerJ Staff

·

Basic reporting

An optic disc and cup segmentation method for glaucoma detection with the application of an attention U-Net incorporating residual mechanism.

The work under review concerns the implementation of neural networks for detecting glaucoma. The topic is interesting. I have some comments on the paper such as:
- There are typo mistakes. Please, edit the work carefully, e.g., ‘… the medical field, people began to …’. There are wrong prepositions and non-long readable sentences, and many parts of the text are repeated multiple times such as ‘In this paper, a glaucoma detection segmentation network model for optic disc and cup disc based on U-Net is proposed,’ etc.
- Correct all syntactic and semantic mismatches in the work.
- References are outdated. More references from the last 3 years could help position the work in research activities.
- The description of the model and results is not convincing. More details and comments on the accuracy are necessary.
- The quality of the pictures is low.
- Figures and tables are not well explained in the text.
- There might be a copyright issue with Figures 6 and 7 [20].

Experimental design

The experimental design is not explained sufficiently.

Validity of the findings

The validity of the findings is good. However, there is no way details on the research to be seen.

Additional comments

The work is not readable. The style, and language need significant improvement.

·

Basic reporting

English language must be edited so it answers the style standards for scientific paper. For example:
"people began to try to use deep learning methods"

Names of programming language Python an library PyTorch are written in lowercase letters, etc.

$\times$ symbol is mistyped.

The abstract and introduction give understandable outline of the topic. However, it does not contain a motivation for the model selection for the proposed solution.

The structure of the publication is with respect to Peerj standards.

Figures 2, 3, 8 and 10 are blurry, the font is small and the contained text is not readable.

Raw data.

Authors provide link to their implementation in Python. Construction of a convolutional neural network using Python / PyTorch is a straightforward task. It is just the usage of existing software.

Experimental design

The experiments described in the text lack formal measures for recognition accuracy, such as precision and recall, for example. This makes hard to evaluate formally the effectiveness of the method for this particular task. Visual analysis is subjective, and it cannot be accepted as proof for performance. The results in Tables 2 and 3 are not completely discussed in the text.

Validity of the findings

"Traditional machine learning methods are far less efficient than deep learning-based image segmentation methods because they require very rigorous formulas, human intervention, and excellent expertise."

Why we could not consider deep learning as a traditional ML method? It has been used for decades already. Also, in many situations other methods perform much better than neural methods.

The paper does not contain any proof why such a heavy model such as deep neural network has to e adopted for a basic image segmentation problem that probably can be loved efficiently using standard image processing techniques.

Additional comments

The topic of the paper is current and it is within the scope of the journal. However, the proposed solution is too heavy for the problem being described. The authors must provide a proof why the tool of neural networks has to be adopted at all.

---

## Round 0.2 · Major Revisions

Please consider the comments of the reviewers.

·

Basic reporting

The work under review concerns the implementation of neural networks for detecting glaucoma. The work is interesting but not so easy to read.

Experimental design

The experiments are provided and explained in the text.

Validity of the findings

It looks like the findings are interesting for the audience,

Additional comments

I still have some comments related to:
- There might be a copyright issue with Figure 2 [10].
- Part of the figures and tables are not cited in the text.
- Use “U-Net” everywhere.
- Edit repetitions such as ‘With the deeper and deeper integration of deep learning and the medical field, people began to try to use deep learning methods to achieve glaucoma diagnosis, which is of great value to areas with glaucoma patients, especially areas lacking ophthalmologists [6-7]. ’ There are many sentences like this one.

·

Basic reporting

English language. Generally correct. However the text contains qualifications whose style does not fit the style of a scientific paper, like "excellent network models".

Intro and background. The statement in the abstract "However, training a professional glaucoma diagnostician is expensive and inefficient, making it impossible to maintain concentration during long hours of work, resulting in a decrease in diagnostic accuracy." can be applied to any medical problem, and suggests that medical specialists have to be replaces by algorithms and machines, which is quite naive. The purpose of methods, as the one proposed in the paper must be to aid medical specialists, not to replace them.

"With the development of artificial intelligence, thinking science research and computer technology, image processing has also become an important content of artificial intelligence research."

-- What do authors classify as "thinking science"?
-- Image processing is a separate field in computer science that adopts methods from AI, but also uses many other approaches that have nothing to do with AI.

"divide the pixels with similar properties in the image into a category and distinguish the pixels belonging to different classifications to enhance the feature parts we need most"

-- this is not a correct definition of image segmentation process. Image segmentation method locates objects' boundaries and contours, and partitions the image into regions defines as sets of pixels.

I cannot agree with authors that AI based segmentation techniques overperform classical computer vision segmentation techniques. For such statement, concrete problems have to be pointed out. In numerous applications, classical algorithms for image segmentation are much faster and more accurate than AI based methods.

As in my previous review, the text contains mistyped characters.

Structure with respect to PeerJ standards. Generally, the structure of the text is within the journal standards.

Figures and tables are clear and readable.

Raw data. The authors provide the source code used in the research written in Python.

Experimental design

Impact and novelty cannot be evaluated from the presented text.

The dataset used in the research is appropriate and it is statistically meaningful, however it is not created by the authors. Something more, one of the terms of the usage of the data set is to cite correctly its origin by "Provided by Medical Image Processing (MIP) group, IIIT Hyderabad", which authors did not include in their proposal.

Validity of the findings

The research is within the scope of the journal. The originality of the research is not obvious from the text of the article. The text does not contain a clear description of the method used in the research.

Research questions are clearly defined. However, many of the arguments stated cannot be accepted, like for example that the AI approaches generally overperform other methods for image segmentation.

Technical standard of investigation cannot be evaluated from the text. It would be impossible to reproduce similar research from just reading the text.

Details of the description of the methods is not enough.

---

## Round 0.3 · Major Revisions

As noted by R2, there are still several issues from their previous comments which must still be addressed.

·

Basic reporting

This work is interesting and implements image processing for glaucoma detection.

Experimental design

The work is explained well.

Validity of the findings

The validity of the findings could be reported in a better way.

Additional comments

There are many figures. The work could be done more compactly.

·

Basic reporting

Line 56: "Segmentation methods based on traditional or machine learning require rigorous mathematical formulas for feature extraction, while image segmentation based on deep learning can independently complete feature extraction according to the label provided, which greatly improves efficiency."

I cannot agree a statement like that to be included into a scientific paper. Complex mathematical structures are indivisible part of the work of a computer scientist. Moreover, convolutional neural networks incorporate an optimization process which is given by mathematical expressions that also can be classified as complex. The structure of the network also is a complex graph. Of course, if you use the mechanism of the neural networks as a black box, you do not focus on the underlying mathematical mechanism.

The mathematical notation used in the paper is not clear and it is ambiguous. After converting to PDF some of the mathematical symbols are misinterpreted, as I have pointed out in my previous two reviews.

Experimental design

I will stay with my previous opinion.

Validity of the findings

I will stay with my previous opinion.

---

## Round 0.4 · Major Revisions

Please accurately consider the comments from Reviewer 2

·

Basic reporting

This work concerns the optic dick and cup segmentation in the detection of glaucoma. It is interesting.

Experimental design

However, there are still small details that need to be improved such as:

- The model is well known. The implementation is a matter of interest. However, the results are spread among multiple tables.

Validity of the findings

Make the work more compact in the first parts and more explanatory in the second parts.

Additional comments

- The use of abbreviations in the titles is not acceptable. Look at OD.
- The work is cut into sections like a book chapter rather than a journal paper.
- Reproducing images from sources [10], [21] may need special permission.

·

Basic reporting

### English language

Has to be edited to fit the style of a scientific text. For example line 77:

> Many people have started using segmentation...

### Intro, background and references

The abstract clearly states the purpose of the text -- development of a neural network for medical images segmentation. The first keyword must not be *glaucoma*, since the paper itself does not represent a scientific research for glaucoma: it is just the application of the presented method.

The introduction clearly explains the importance of the application: glaucoma detection and diagnosis. However, the part that contains the introduction on the topic of image processing and segmentation can be significantly improved.

Line 38:
> Image segmentation is one of the key technologies of image processing.

Image segmentation is not a technology. It is a set of methods and algorithms that are part of image processing and image preprocessing.

> It mainly locates the boundaries and contours of the target in the form of regions, > and the computer must understand the image.

Image segmentation is not focused only on the contour detection of the objects in the image. For example there are algorithms that segment set of pixels based on the color information in the image. Computers do not understand anything -- the are just machines.

> Traditional machine learning is sensitive to the noise and complexity of images.

The same statement fully holds for neural networks as well. Also there are "traditional" image processing techniques, which are quite robust against noise: for example Hough transform in geometrical shapes detection.

### Text structure

Sections have too much short subsections. The text still contains mistyped mathematical symbols.

### Figures and tables

Some images still contain blurry information.

### Raw data

Authors provide their method written in Python.

Experimental design

### Originality of the research and scope of the journal

The topic of the text is within the scope of the journal. There are many papers published for neural network design for diagnosis of glaucoma. The proposed text is focused only on the segmentation process.

### Research questions definition

Research questions are clearly defined.

Validity of the findings

### Impact and novelty

There are two major drawbacks:

1. The text does not contain any real proof that the segmentation technique based on deep learning performs better than (as the authors call them) "traditional methods".
2. The text does not contain any real proof that the technique presented here performs better than the other neural networks in the literature that focus on diagnosis of glaucoma.

Additional comments

There are many papers published on the topic that cover the whole process for glaucoma detection using neural networks. The proposed work covers the segmentation part of this process, and still it is not clear from the text why the proposed method is unique. The parts of the text that are focused on image processing need serious editing.

---

## Round 0.5 · Minor Revisions

Based on the Reviewer 1, I recommend minor revisions to improve the explanation of the results and the quality of figures.

·

Basic reporting

The work under review is interesting and concerns an important topic on optic disks and cup segmentation for glaucoma detection.

Experimental design

The experiments are well presented. Some more explanation of the results may improve the quality of the work.

Validity of the findings

Some more explanation of the results may improve the quality of the work.

Additional comments

The work is readable and well structured.
I would recommend improving the quality of the figures.

·

Basic reporting

I am extremely sorry, but I do not see any of the important remarks to be fixed in this version of the proposition, and I have to reject it.

Experimental design

n/a

Validity of the findings

n/a

Additional comments

n/a

---

## Round 0.6 · accepted · Accept

Based on the issues raised, the manuscript is correctly reworked, and it can be accepted.